# Microbial Lipopeptide-Producing Strains and Their Metabolic Roles under Anaerobic Conditions

**DOI:** 10.3390/microorganisms9102030

**Published:** 2021-09-25

**Authors:** Jia-Yi Li, Lu Wang, Yi-Fan Liu, Lei Zhou, Hong-Ze Gang, Jin-Feng Liu, Shi-Zhong Yang, Bo-Zhong Mu

**Affiliations:** 1State Key Laboratory of Bioreactor Engineering, School of Chemistry and Molecular Engineering, East China University of Science and Technology, Shanghai 200237, China; Y30190306@mail.ecust.edu.cn (J.-Y.L.); liuyifan@ecust.edu.cn (Y.-F.L.); leizhou@ecust.edu.cn (L.Z.); ganghz@ecust.edu.cn (H.-Z.G.); ljf@ecust.edu.cn (J.-F.L.); bzmu@ecust.edu.cn (B.-Z.M.); 2State Key Laboratory of Enhanced Oil Recovery, Research Institute of Petroleum Exploration and Development, CNPN, Beijing 100083, China; luwangmoon@petrochina.com.cn; 3Engineering Research Center of MEOR, East China University of Science and Technology, Shanghai 200237, China

**Keywords:** biosurfactant, anaerobic bacteria, nitrate respiration, non-ribosomal peptide synthetase (NRPSs)

## Abstract

The lipopeptide produced by microorganisms is one of the representative biosurfactants and is characterized as a series of structural analogues of different families. Thirty-four families covering about 300 lipopeptide compounds have been reported in the last decades, and most of the reported lipopeptides produced by microorganisms were under aerobic conditions. The lipopeptide-producing strains under anaerobic conditions have attracted much attention from both the academic and industrial communities, due to the needs and the challenge of their applications in anaerobic environments, such as in oil reservoirs and in microbial enhanced oil recovery (MEOR). In this review, the fifty-eight reported bacterial strains, mostly isolated from oil reservoirs and dominated by the species *Bacillus subtilis*, producing lipopeptide biosurfactants, and the species *Pseudomonas aeruginosa*, producing glycolipid biosurfactants under anaerobic conditions were summarized. The metabolic pathway and the non-ribosomal peptide synthetases (NRPSs) of the strain *Bacillus subtilis* under anaerobic conditions were analyzed, which is expected to better understand the key mechanisms of the growth and production of lipopeptide biosurfactants of such kind of bacteria under anaerobic conditions, and to expand the industrial application of anaerobic biosurfactant-producing bacteria.

## 1. Introduction

Biosurfactants are a group of microbial secondary metabolites with strong surface/ interfacial activity mainly produced by bacteria, yeasts, and fungi [1], and a kind of amphiphilic compounds with a wide structural variety including lipopeptide, glycolipid, phospholipid, polysaccharide-protein complexes, fatty acids, or natural lipids, etc. [2]. The microbial lipopeptide is one of the representative biosurfactants and is characterized as a series of structural analogues of different families. Thirty-four families covering about 300 lipopeptide compounds have been reported in the last decades, and among those families the surfactin, iturin, fengycin, and lichenysin are the most frequently reported ones. Microbial lipopeptides have been widely studied for their diverse biotechnological applications and have served as emulsifiers and stabilizers in food industry [3], formulations in cosmetic industry [4], oil displacement agent in microbial enhanced oil recovery (MEOR) [5], biocontrol agent in agriculture, and biodegradation and bioremediation in environmental protection system [6]. It has been proved that *Bacillus* [7], *Pseudomonas* [8], *cyanobacteria* [9], *actinomycetes* [10,11], and *fungi* [12] can produce lipopeptide biosurfactants. The families of iturin [13] and fengycin [14] were discovered in 1949 and 1986, respectively. Surfactin was first found in the culture medium of *Bacillus subtilis* in 1968, and it is a cyclic (polar) heptapeptide attached to *a β*-OH (lactone) fatty acid chain (11 to 16 carbons) [15,16,17,18]. Different modifications in the chiral sequence of amino acids in the peptide chain can occur; amino acids of the aliphatic group, Val, Leu, and Ile in positions 2nd, 4th, and 7th were already observed. Known for the exceptional emulsion and foamability, surfactin can reduce water surface tension from 72 to 27 mN/m at ≈10 mg/L [19,20]. In addition to the powerful surfactant properties, surfactin showed broad biological activities such as hemolytic activities, modification of the performance of enzymes, and interaction with membranes.

Lipopeptide-producing bacteria have been extensively reported under aerobic conditions. Only a few bacteria have been identified as being able to produce lipopeptides in the absence of oxygen. However, a large number of environments inherited oxygen free conditions, such as deep sea and underground oil reservoirs, and the application of anaerobic lipopeptide-producing bacteria in industries is still a challenge. This review presented a summary on the fifty-eight reported bacterial strains that can produce biosurfactants under anaerobic conditions. The metabolic pathway and the non-ribosomal peptide synthetases (NRPSs) of the strain *Bacillus subtilis* under anaerobic conditions were analyzed, which is expected to lead to better understanding of the key mechanisms of the growth and production of lipopeptide biosurfactants of such kind of bacteria under oxygen-limiting conditions, and to expand the biotechnological and industrial applications of anaerobic lipopeptide-producing bacteria.

## 2. Lipopeptide-Producing Strains under Anaerobic Conditions

According to their tolerance to oxygen, the lipopeptide-producing microorganisms can be roughly divided into five categories: obligate aerobes, microaerophies, facultative anaerobes, aerotolerant anaerobes, and obligate anaerobes [21]. Most of the reported anaerobic microorganisms have been bacteria and archaea until now, as well as a small number of actinomycetes, mycoplasma, and fungi [22,23]. Since 1980, Cooper et al. discovered that *Clostridium pasteurianum* produced extracellular neutral lipopeptide surfactant under anaerobic conditions, which could reduce the surface tension of deionized water from 72 mN/m to 55 mN/m [24]. Up to now, there have been about thirty-six strains reported that can produce lipopeptide biosurfactant under anaerobic conditions, mainly isolated from oil reservoirs and oil contaminated soil samples, and dominated by the genera *Bacillus, Pseudomonas*, and *Yeast**s*. There were also reports from lake sediments and hot spring samples. Among them, the lipopeptide surfactant produced by *Bacillus mojavensis* GMTB-C1-2 screened from submarine oil reservoirs showed a good surface activity under anaerobic conditions, which could reduce the surface tension of deionized water from 72 mN/m to 27 mN/m (Ghojavand et al., 2011) [25]. *Bacillus licheniformis* WJ-2, which was isolated from in Daqing Oilfield, China, in 2012, showed the best lipopeptide yield of 1.69 g/L under anaerobic conditions [26]. The information of lipopeptide-producing strains (*Bacillus*) is summarized in Table 1.

In addition to *Bacillus*, a few of the lipopeptides-producing bacteria belonging to the *Pseudomonas, Pallidobacterium, Rhodobacteria*, and *Piperacilla* were also reported. In 2012, *Geobacillus pallidus H9* was isolated from Daqing Oilfield, China, and the yield of macromolecular polysaccharide protein complex was up to 2.8 g/L under anaerobic conditions at 65 °C [27]. In addition, it has been reported that *Anaerophaga thermohalophila Fru22T* (Oilfield, Hannover, Germany) [28], *Rhodococcus ruber Z25* (Daqing Oilfield, Heilongjiang, China) [29], *Tepidibacter mesophilus B1T* (Karamay Oilfield, Xinjiang, China) [30], and *Luteimonas huabeiensis HB-2* (Baogeli Oilfield, Xinjiang, China) [31] can produce lipopeptides under anaerobic conditions. The earliest mention of production of biosurfactants under anaerobic conditions was published in 1955 [32], where another twenty-two strains of anaerobic wild bacteria producing glycolipid biosurfactants or other biosurfactants were reported. The information around other biosurfactant-producing strains is summarized in Table 2.

**Table 1 microorganisms-09-02030-t001:** Information of lipopeptide-producing strains (*Bacillus*) under anaerobic conditions.

Year	Source	Strain	Biosurfactant	References
1980	Unknown	*Clostridium pasteurianum*	Neutral lipid	[24]
1985	Oklahoma oilfield, US	*Bacillus mojavensis* JF-2	Lipopeptide	[33]
1995	Oilfield, North Germany	*Bacillus licheniformis* BAS50	Lichenysin A	[34]
1997	Oilfield, North Germany	*Bacillus licheniformis* BNP29	Lipopeptide	[35]
1997	Oilfield, North Germany	*Bacillus licheniformis* BNP36	Lipopeptide	[35]
1997	Oilfield, North Germany	*Bacillus licheniformis* Mep132	Lipopeptide	[35]
1997	Oilfield, Russian	*Bacillus subtilis* C9	Surfactin	[36]
1999	Marine sediments	*Bacillus subtilis* ATCC 21332	Lipopeptide	[37]
2000	Noyabrskyi oilfield, Russian	*Bacillus subtilis* BS2202	Lipopeptide	[38]
2001	Water buffalo	*Bacillus licheniformis* 26 L-10	Lipopeptide	[39]
2004	Oklahoma oilfield, US	*Bacillus subtilis* ATCC 12332	Lipopeptide	[40]
2007	Sahara desert, Tunisia	*Bacillus subtilis* RS-1	Lipopeptide	[41]
2007	Oilfield, Iran	*Bacillus subtilis* PTCC 1365	Lipopeptide	[42]
2008	Unknown	*Bacillus subtilis* ATCC6633	Mycosubtilin, surfactin	[43]
2009	Russia	*Bacillus licheniformis* VKM B-511	Licheniformin A	[44]
2009	Hot springs, US	*Bacillus licheniformis* TT33	Lipopeptide	[45]
2011	Oilfield, Iran	*Bacillus mojavensis* GMTB-C1-2	Lipopeptide	[25]
2011	Oilfield, US	*Bacillus cereus* ATCC14579	Lipopeptide	[46]
2012	Oilfield, Brazil	*Bacillus subtilis* 309	Sufactin	[47]
2012	Oilfield, Brazil	*Bacillus subtilis* 191	Lipopeptide	[47]
2012	Oilfield, Brazil	*Bacillus subtilis* 311	Sufactin	[47]
2012	Oilfield, Brazil	*Bacillus subtilis* 552	Lipopeptide	[47]
2012	Oilfield, Brazil	*Bacillus subtilis* 573	Sufactin	[47]
2012	Daqing oilfield, China	*Bacillus licheniformis* WJ-2	Glycosides, Lipopeptides	[26]
2012	Soil from Ituri, Congo	*Bacillus**amyloliquefaciens* S499	Surfatin, Iturin and Fengycin	[48]
2015	Oilfield, Germany	*Bacillus subtilis* DSM 10T	Lipopeptide	[49]
2017	Daqing oilfield, China	*Bacillus licheniformis* DQ4	Lipopeptide	[50]
2017	Xinjiang oilfield, China	*Bacillus amyloliquefaciens* 702	Glycosides, Lipopeptides	[51]
2018	Shengli oilfield, China	*Bacillus licheniformis*	Glycosides, Lipopeptides	[52]
2019	Shengli oilfield, China	*Bacillus tequilensis*	Glycosides, Lipopeptides	[53]
2021	Xinjiang oilfield, China	*Bacillus subtilis* AnPL-1	Sufactin	[54]

These biosurfactant-producing bacteria mostly choose nitrate or sulfate as the alternative electron acceptor under oxygen deprivation. Generally, these studies presented higher lipopeptide yields under oxygen-rich conditions than in oxygen limiting conditions. It is speculated that their production is reduced under anaerobic conditions, considering that the available energy is preferentially used in primary metabolic pathways rather than secondary metabolic pathways. The slow growth of bacteria in oxygen-limiting conditions can also reduce the growth-dependent products of lipopeptide. On the other hand, the difference of intercell microenvironments under aerobic and anerobic conditions may affect the substrate preferences change due to the redox potential of the biochemical reactions and the usage of different electron acceptors [53]. This may lead to changes in the metabolic pathways involved in the production of lipopeptide. When bacteria encounter the transformation from oxygen enriched environment to hypoxia environment, it is of great significance for bacteria to make a reasonable stress transformation for their growth and metabolism. However, there are few reports on the differences of biosurfactant secondary metabolism of these bacteria under aerobic and anaerobic conditions.

**Table 2 microorganisms-09-02030-t002:** Information of biosurfactant-producing strains under anaerobic conditions.

Year	Source	Strain	Biosurfactant	References
1955	Ain-ez-Zania lake, Lybia	*Desulfovibrio desulfuricans* sp. *DSM 1926*	Unidentified	[32]
1991	Hot springs, USA	*Thermoanaerobacter pseudethanolicus ATCC 33233*	Unidentified	[55]
2000	Petroleum contaminated soil	*Pseudomonas* sp.BS2201	Unidentified	[56]
2000	Petroleum contaminated soil	*Pseudomonas* sp. BS2203	Unidentified	[56]
2000	Oilfield, India	*Pseudomonas aeruginosa* ATCC 10145	Rhamnolipid	[57]
2002	Oilfield, Germany	*Isolate* Glc2	Unidentified	[58]
2007	Antarctic soil	*Pantoa* A-13	Rhamnolipid	[59]
2007	Hospital Wastewater	*Pseudomonas aeruginosa* PAO1	Rhamnolipid	[60]
2008	Shengli oilfield, China	*Pseudomonas aeruginosa* SH6	Rhamnolipid	[61]
2010	Municipal Sewage Sludge	*Pseudomonas aeruginosa* ANBIOSURF-1	Rhamnolipid	[62]
2012	Oilfield; Veracruz, Mexico	*Thermoanaerobacter* sp.	Unidentified	[63]
2012	Menggulin oilfield, China	*Pseudomonas aeruginosa* WJ-1	Rhamnolipid	[64]
2013	Soil samples, Iowa, USA	*Pseudomonas aeruginosa* E03-40	Rhamnolipid	[65]
2013	Oil contaminated soil, Iran	*Enterobacter cloacae*	Unidentified	[66]
2014	Gachsaran oilfield, Iran	*Bacillus**stearothermophilus* SUCPM#14	Unidentified	[67]
2015	Xinjiang oilfield, China	*Pseudomonas aeruginosa* SG	Rhamnolipid	[68,69,70]
2017	Xinjiang oilfield, China	*Pseudomonas aeruginosa* 709	Rhamnolipid	[60]
2018	Daqing oilfield, China	*Pseudomonas aeruginosa* DQ3	Rhamnolipid	[71]
2018	Daqing oilfield, China	*Pseudomonas aeruginosa* DQ1	Rhamnolipid	[71]
2018	Daqing oilfield, China	*Pseudomonas aeruginosa* DQ5	Rhamnolipid	[71]
2018	Daqing oilfield, China	*Pseudomonas aeruginosa* DQ6	Rhamnolipid	[71]
2020	Dagang oilfield, China	*Bacillus licheniformis* DM-1	Exopolysaccharide	[72]

## 3. The Mechanism of Anaerobic Growth of *Bacillus subtilis*

According to the respiration mode of microorganisms, it can be divided into aerobic respiration, anaerobic respiration, and fermentation [73]. During aerobic respiration, electron transfer phosphorylation uses oxygen as an electron acceptor. Inorganic and organic compounds, such as nitrate and fumarate, are used as alternative electron acceptors in electron transfer pathway during anaerobic respiration; in anaerobic fermentation, energy is phosphorylated at the substrate-level, and electrons are transferred to the intermediate metabolite receptor molecule, rather than electron transfer phosphorylation to generate energy [74,75]. *B. subtilis*, a strict aerobe that has been proved to be a facultative anaerobe, can use nitrate as terminal electron acceptor for anaerobic respiration, and can also carry out mixed-acid/butanediol anaerobic fermentation in the presence of glucose, pyruvic acid, or amino acid [76]. The fermentation products include ethanol, acetic acid, lactic acid, and acetone [77].

### 3.1. Anaerobic Regulatory Network of Bacillus subtilis

A complete set of regulatory networks of *Bacillus subtilis* was stimulated by oxygen level change to make stress adjustments [78]. Gene expression related to anaerobic nitrate respiration and anaerobic pyruvate fermentation process is activated by the membrane-bound ResD-ResE two-component signal transduction system, the anaerobic regulator Fnr, which is activated by the changes in environmental oxygen levels, and the redox-sensing repressor Rex [79]. As shown in Figure 1, the sensor histidine kinase ResE is autophosphorylated when it senses the limitation of available oxygen, and then provides high-energy phosphate group to the homologous reaction regulator ResD, which makes ResD phosphorylation into ResD-P [80]. The expression of *ldhlctP* operon (encoding L-lactate dehydrogenase and lactate permease), *nasDEF* gene (encoding the assimilatory nitrite reductase), *hmp* gene (coding for a flavohemoglobin), and *ctaA* gene (encoding a heme A sythase) are regulated by ResDE [81]. Meanwhile, the genes of nitrate respiration and anaerobic fermentation are induced by the total anaerobic regulator ResD-ResE.

*B. subtilis* has encoded two distinct nitrate reductases, one for the assimilation of nitrate nitrogen and the other for nitrate respiration, both of which are induced by the anaerobic transcriptional regulator Fnr [82]. Fnr, a member of the catabolite gene activator protein (CAP) family of transcriptional regulators, induces the expression of nitrate respiration genes with absence of oxygen. The transcriptional units (the nitrate reductase encoded by *narGHJI* operon, nitrite transporters encoded by *narK* and *fnr* gene itself) directly activated by the anaerobic regulator (Fnr) under the oxygen limitation and the presence of nitrate. The *arfM* gene, encoding anaerobic respiration and a fermentation regulator, also belongs to Fnr-bingding promoter of the regulon genes [83]. 

Rex, a redox-sensing repressor of *B. subtilis*, responds to the change of NADH/NAD^+^ ratio, which is related to the oxygen concentration within individual microenvironment. ResED activated the anaerobic transcription of *ldhlctP, cydABCD*, and *ywcJ* operons [84]. NADH binds with higher affinity to Rex than NAD^+^, inducing a domain rearrangement followed by the release of the repressor from the promoter. When the NADH/NAD^+^ ratio increases, the rearranged Rex dimer is very inefficient to repress the expression of anaerobic fermentation genes. Recycling of NADH is accomplished by conversion of pyruvate to fermentation products [85,86].

In the anaerobic fermentation of *B. subtilis*, two molecules of pyruvate are concentrated into acetolactate by acetolactate synthase (ALSS); Acetolactate is then converted to acetoacetate by the acetolactate decarboxylase ALSD (both enzymes are encoded by the *alsSD* operon) [87]. The production and secretion of acetoacetic acid is a mechanism for bacteria to maintain a constant pH value in individual microenvironment. The acidic medium of *Bacillus licheniformis* and *Bacillus subtilis* can induce the expression of *alsSD* operon, while the anaerobic and stationary phase expression of *alsSD* is *alsR*-controlled gene [88,89]. Acetic acid or acidic pH in the medium induced *alsR* gene to express transcription regulator AlsR, which shows significant homology to the LysR family of bacterial activator proteins [90]. The *alsR* gene is located in the upstream of *alsSD* operon, which effectively activates the transcription of *alsSD* operon, thus promoting acetoin synthesis in the mixed-acid/butanediol fermentation of *B. subtil* is [91,92].

### 3.2. Anaerobic Energy Metabolism of Bacillus subtilis

Bacteria switches between two kinds of energy generation in response to environmental changes: one is the generation of ATP at the substrate-level phosphorylation by chemical bond cleavage or oxidation, the other is the energy synthesized by the proton concentration gradient difference between inside and outside the cell, which is driven by the redox potential difference in the process of electron transfer [93,94]. In this process, oxygen and some inorganic salts (nitrate, N-oxide, dimethyl sulfoxide (DMSO), fumarate, Fe (Ш), Mn (IV), sulfate, and many other compounds) served as external electron acceptors receive electrons and maintain the oxidation and reduction potential equilibrium in cells [81]. NADH + H^+^ and NAD^+^ act as electron donor and electron acceptor in respiratory electron transfer chain to maintain intracellular potential balance, respectively. In the case of sufficient oxygen, NADH + H^+^ is oxidized to NAD^+^ through tricarboxylic acid cycle. When oxygen is deficient, the ratio of NADH + H^+^/NAD^+^ is unbalanced, so bacteria have to choose other strategies such as nitrate respiration or pyruvate fermentation to complete the reoxidation of NADH + H^+^ [95]. The process of nitrate respiration and anaerobic fermentation is depicted in Figure 2.

Unlike *Escherichia coli* and *Bacillus licheniformis*, the *Bacillus subtilis* does not grow anaerobically in glycerol succinate or fumarate medium because it lacks glycerol-3-dehydrogenase gene and fumarate reductase gene, and only has limited primary electron donor dehydrogenase, namely various forms of NDH-II and aerobic glycerol-3-phosphate dehydrogenase [96]. The nitrate reductase system of bacteria covered three types in *E. coli*: cytoplasmic assimilative NAD(P)H-dependent nitrite reductases Nas, encoded by *nasDE*, membrane-bound respiratory nitrate reductases Nar, encoded by *narGHJI*, and the periplasmic dissimilatory nitrite/nitrate reductases Nap, encoded by the *napFDAGHBC* operon. These three kinds of proteins belong to the dimethyl sulfoxide reductase (DMSO) family and contain the *bis*-molybdopterin guanine dinucleotide (MGD) cofactor [97]. However, only two nitrate reductases function in nitrogen assimilation and in respiratory play a role in the process of nitrate respiration of *B. subtilis* [98]. In the electron transfer chain of nitrate respiration, two electrons flow through the heme *b* of the cytochrome *b* subunit (NarI) to the iron-sulfur clusters of the soluble subunit (NarH), and finally to the cytoplasmic subunit (NarG). NarG reduces nitrate to nitrite. Due to the different sites of quinol oxidation and nitrate reduction in the cytosol, nitrate reductase Nar contributes to the production of proton gradient. Nitrite is then further converted to ammonium by assimilatory nitrite reductase NasDE [99].

In the absence of external electron acceptors, pyruvate was transformed into lactic acid, acetone, 2,3-butanediol, ethanol, and acetic acid during anaerobic fermentation [49]. The most important thing in fermentation is the reoxidation of intracellular NADH+H^+^. NAD^+^ regeneration is mainly mediated by cytoplasmic lactate dehydrogenase (encoded by *ldh*) by converting pyruvate to lactate [100]. Acetate is produced from acetyl-CoA through two consecutive reactions catalyzed by phosphotransaminase and acetate kinase (encoded by *pta* and *ackA* operons), and ATP is generated in the process. Acetone can be converted to pyruvate, which is catalyzed by acetolactate synthase and acetolactate decarboxylase (encoded by *alsSD*), and then ethylene ketone is then reduced to 2,3-butanediol by 2,3-butanediol dehydrogenase (encoded by *bdhA*), with NAD^+^ regeneration. During fermentation, ethanol is produced by acetyl-CoA, which first converts acetaldehyde to ethanol through acetaldehyde dehydrogenase [101].

## 4. Non-Ribosomal Peptide Synthetases (NRPSs) of Surfactin

Quorum Sensing (QS) is a type of population density-dependent cell–cell signaling that triggers changes in behavior when the population reaches a critical density. Quorum sensing systems rely on the production and sensing of extracellular signals [80]. In the gram-positive *B. subtilis*, the genetic competence (the ability to absorb foreign DNA from the environment), sporulation, the production of degrading enzymes and extracellular polysaccharides, and the synthesis of surfactin are all regulated by quorum sensing system [102]. This article describes the NRPSs mechanism of surfactin biosynthesis under aerobic conditions and the effect of correlation signal molecules on synthesis of surfactin in *B. subtilis*. In *B. subtilis,* surfactin is mainly synthesized by non-ribosomal peptide synthetases (NRPSs) which are exceptional megaenzymes that have evolved in bacteria and fungi to assemble highly complex, bioactive secondary metabolites of peptide origin [103,104]. To perform peptide complex chemical assembly, NRPSs rely on an array of large, repetitive catalytic units called modules, each comprised of several catalytic domains covalently linked within a single polypeptide chain [105]. At present, many factors affecting surfactin biosynthesis under aerobic conditions have been reported, but there are few reports about the factors influencing surfactin biosynthesis under anaerobic conditions.

### 4.1. Regulator of Surfactin Synthesis in Bacillus subtilis

The transcription and expression of surfactin synthesis *srfA* gene are regulated by a large number of regulatory factors, the most important is the two-component signal transduction system (TCS) in *B. subtilis*. Two-component signal transduction systems are a prototypical signaling cascade that are used by bacteria to couple changes in the extracellular environment to physiological effects [105,106]. Typically, TCS comprise a sensor histidine kinase (HK), which consists of an input domain, detecting a signal, and a kinase domain, and a response regulator (RR). The sensor histidine kinase after the change of external environment is detected, the phosphorylation of the sensor’s histidine kinase will transfer to the conserved aspartic residue on the N-terminal receptor domain of the response regulator, resulting in the conformational change of the receptor, thus completing the signal transmission. *B. subtilis 168* encodes for about 36 histidine kinases and 34 response regulators [107]. Here we will describe three TCS that are highly correlated with *srfA* operon transcription and expression: the ComAPQSX system, the DegS-DegU system, and the Rap-Phr system (Figure 3).

In the two-component signal transduction system of ComAPQSX, the modified extracellular peptide pheromone ComX interacts with the sensor histidine kinase ComP, which is autophosphorylated after stimulation and then transferred to the serine residue of ComA. Phosphorylated ComA binds to the ComA box upstream of *srfA* promoter (T/GCGG-N_4_-CCGCA) in the form of tetramer and initiates the transcription of *srfA* operon [108,109]. Secondly, the expression of *srfA* was repressed after hydrogen peroxide (H_2_O_2_) treatment. PerR [110], a dimeric zinc protein with a regulatory site that coordinates either a Fe^2+^ or a Mn^2+^ metal ion, actively regulates the expression of *srfA* by binding to the PerR box in the upstream region of ComA box, and H_2_O_2_ inhibits the binding activity of PerR to DNA [111].

Another transcription factor, the DegS-DegU system, controls transcription of *srfA* genes in a manner depending on the level of the phosphorylated response regulator. The DegS-DegU system regulates many cellular processes, including exoprotease production, motility, biofilm formation, *γ*-polyglutamic acid production, and competence development in *B. subtilis.* The DegS autophosphorylates after sensing the signal of environmental changes, then the phosphoryl group is transferred to a conserved aspartic acid residue on the N-terminal receiver domain of the DegU. The phosphorylated DegU can effectively stimulate the transcription of *srfA* [105,112].

Eleven aspartate phosphatase proteins (RapA-RapK) are encoded by *B. subtilis*, among which RapC, RapF, RapG, RapH, and RapK are negative regulators of *srfA* transcription by inhibiting the binding of phosphorylated ComA-P to *srfA* promoter DNA [113]. Eight Phr peptides (PhrA, PhrC, PhrE, PhrF, PhrG, PhrH, PhrI, and PhrK) are encoded in *B. subtilis*. *phr* gene, which is located downstream of *rap* operon, and each Phr peptide inhibits the activity of Co-transcriptional Rap protein. Thus, it indirectly promotes the combination of ComA-P and *srfA* promoter, and plays a positive stimulating role [114]. RghR can inhibit the production of RapDGH protein by inhibiting the transcription of *rapDGH* operon, thus reducing the interference of RapDGH on ComA-P. and eventually promote the transcription of *srfA* operon [115].

By binding to the C-terminal domain of RNA polymerase (RNAP) subunit, Spx blocks the complex formation by preventing ComA-P and RNAP from binding to promoter, thus blocking *srfA* transcriptional activation. In addition, in high concentrations of amino acids such as Ile, Leu, and Val, Cody and AbrB inhibit the specific interaction between ComK and *srfA* promoter, resulting in the down-regulation of *srfA* transcription [116]. High levels of superoxide can specifically inhibit the transcription of *comQXP* operon, while superoxide dismutase SodA can reduce the interference of superoxide, thus promoting the transcriptional activation of *comQXP* operon and indirectly promoting the transcription of *srfA* operon [117]. Secondly, ATP-dependent proteolytic enzyme ClpXP effectively inhibits the binding of the C-terminal domain of RNA polymerase subunit (RNAP) by hydrolyzing RNA polymerase binding protein Spx, and indirectly promotes the formation of ComA-P complex with RNAP [118].

### 4.2. Non-Ribosomal Peptide Sythetases (NRPSs)

The biosynthesis of surfactin is based on non-ribosomal peptide synthetases (NRPSs) [119,120]. In non-ribosomal peptide synthetases (NRPSs) systems, multiple NRPSs subunits interact with each other in a specific linear order mediated by specific docking domains (DDs), to synthesize well-defined peptide products [121]. A simple extension module of NRPSs consists of at least three essential domains in the order of C-A-PCP: an adenylation (A) domain, which is responsible for the selection and activation of substrates to aminoacyl-adienoate. A small peptidyl carrier protein (PCP) domain that carries all acyl-intermediates on the terminal-SH group of its 4′-phosphopantetheine (Ppant) cofactor. A condensation (C) domain that forms peptide bonds between the acyl-S-PCP intermediates of two adjacent modules [122]. In the process of surfactin biosynthesis, the termination module contains an additional thioesterase (TE) domain responsible for the release of the product by hydrolysis or cyclization to form cyclic or ring branched molecules [123]. In addition, other domains are responsible for the modification of the peptide rings, acetylation, glycosylation, and lipidization domains modify the polypeptide skeleton [124,125].

## 5. Conclusions

This review summarizes the thirty-six reported lipopeptide biosurfactants-producing strains and twenty-two other biosurfactants-producing stains, mostly isolated from oil reservoirs. The reported anaerobic biosurfactant-producing bacteria include thirty-seven strains of *Bacillus*, sixteen strains of *Pseudomonas*, one strain of *Clostridium pasteurianum*, two strains of *Thermoanaerobacter*, one *Anaerophaga thermohalophila*, one *Geobacillus pallidus*, one *Rhodococcus ruber*, one *Tepidibacter mesophilus*, one *Luteimonas huabeiensis*, one *Enterobacter cloacae*, and one *Desulfovibrio desulfuricans*. Nitrate is an alternative electron acceptor mainly for strains under oxygen-limiting conditions, including *Desulfovibrio* and *Thermoanaerobacter*, which select sulfate as an alternative electron acceptor. This shows that bacteria are able to produce biosurfactants under anaerobic conditions, although the anaerobic biosurfactant yield is much lower than that under aerobic conditions. The low anaerobic biosurfactant production in bacteria is contributed by the slow growth of bacteria in oxygen-limiting conditions. At the same time, the limited energy, producing in anaerobic conditions, is preferentially used in primary metabolic pathways rather than secondary metabolic pathways, which is also a major factor affecting the yield of biosurfactants. On the other hand, the difference of intercell microenvironments under oxygen-rich and oxygen-limited environments may affect the substrate selection due to the redox potential of the biochemical reactions and the usage of different electron acceptors. In general, the interaction of various factors leads to the low biosurfactants production of bacteria. Thus, understanding the factors affecting biosurfactant-production under microaerophilic and anaerobic conditions is important to the promising applications of bacteria in bioremediation, microbial enhanced oil recovery (MEOR), and the oxygen-limiting environments. Most of the research on microbial enhanced oil recovery (MEOR) by biosurfactant-producing bacteria under anaerobic conditions was in the laboratory stage, and the in-situ microbial flooding experiment on MEOR by anaerobic bacteria was still scarce. It is of great significance for the research and application of anaerobic biosurfactant production technology independent of gas injection to screen the bacteria resource of anaerobic biosurfactant production from the reservoir and analyze its biosurfactant production and metabolism process under the condition of oxygen deficiency. 

In general, during the utilization of any anaerobically produced biosurfactants bacteria for bioremediation or MEOR processes, the nature of used bacteria both from biosurfactants structure and the anaerobic growth and metabolic pathway must be examined first. *Bacillus subtilis* is most important of the anaerobic biosurfactant producer, the biosynthesis of surfactin is based on non-ribosomal peptide synthetases (encoded by *srfA* operon), which is regulated by quorum sensing system, a density-dependent signaling mechanism of microbial cells, and two-component signal transduction system (TCS), which is used to couple changes in the extracellular environment to physiological effects. *Bacillus subtilis* relies on a set of internal complex regulatory network to make adjustments and choose the appropriate generation energy and metabolic pathway to cope with environmental changes. It is proven that the facultative anaerobe *Bacillus subtilis* may use nitrate replace oxygen as terminal electron acceptor to transform energy and maintain the oxidation and reduction potential equilibrium in cell under oxygen-limiting conditions. The pyruvate fermentation is also an option in the absence of available electron acceptor and oxygen. The energy required by *Bacillus subtilis* to synthesize lipopeptide can be obtained by nitrate respiration or pyruvate fermentation in oxygen limited environments. However, how the synthesis process of surfactin will change and adjust when the environmental oxygen level changes, and the specific impact degree and transformation strategy of *Bacillus subtilis*, were unclear. Future research should intensify efforts in the difference of biosurfactants production of bacteria under the aerobic and anaerobic conditions. At the same time, the optimization of anaerobic biosurfactant production is also very effective for expanding the application range of anaerobic biosurfactant-producing bacteria.

## Figures and Tables

**Figure 1 microorganisms-09-02030-f001:**
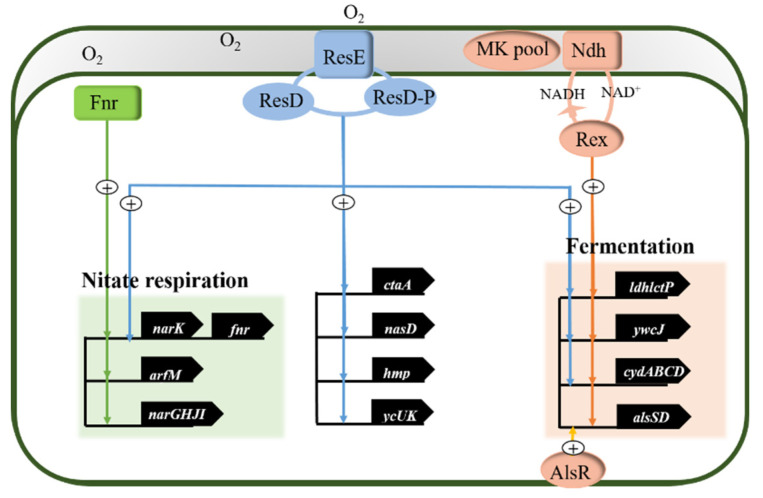
Anaerobic regulation network of *Bacillus subtilis*. Adapted from [81]. The green line represents that the Fnr regulator positively induces the nitrate respiration genes expression including *narK* gene, *fnr* gene, *arfM* gene, and *narGHJI* operon; The blue line represents that the ResD-ResE system positively regulate the expression of *ldhlctP* operon (encoding L-lactate dehydrogenase and lactate permease), *nasDEF* gene (encoding the assimilatory nitrite reductase), *hmp* gene (coding for a flavohemoglobin), *ctaA* gene (encoding a heme A sythase), *narK* gene (encoding the nitrate reductase), *fnr* gene (encoding the Fnr regulator), *ywcJ* gene (encoding a heme A sythase), and *cydABCD* gene (encoding a heme A sythase); The orange line represents that the Rex regulator positively stimulates the fermentation genes expression, including *ldhlctP* operon, *ywcJ* gene, *cydABCD* gene, and *alsSD* gene. The AlsR regulator positively induces the *alsSD* gene expression.

**Figure 2 microorganisms-09-02030-f002:**
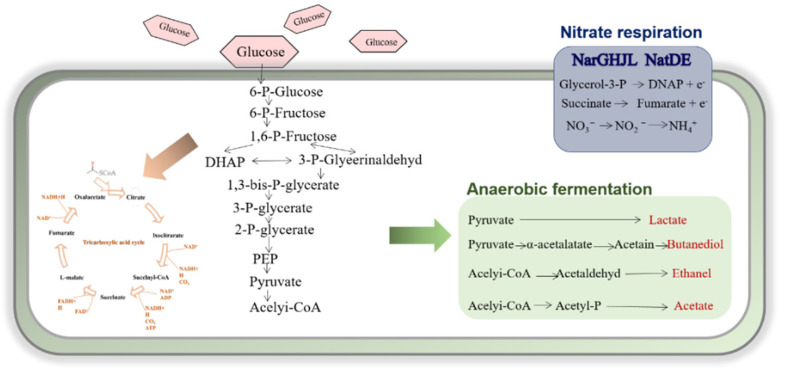
Glycolysis pathway and nitrate respiration pathway of *Bacillus subtilis*. Regulation of the anaerobic metabolism in *Bacillus subtilis.* Adapted from [81].

**Figure 3 microorganisms-09-02030-f003:**
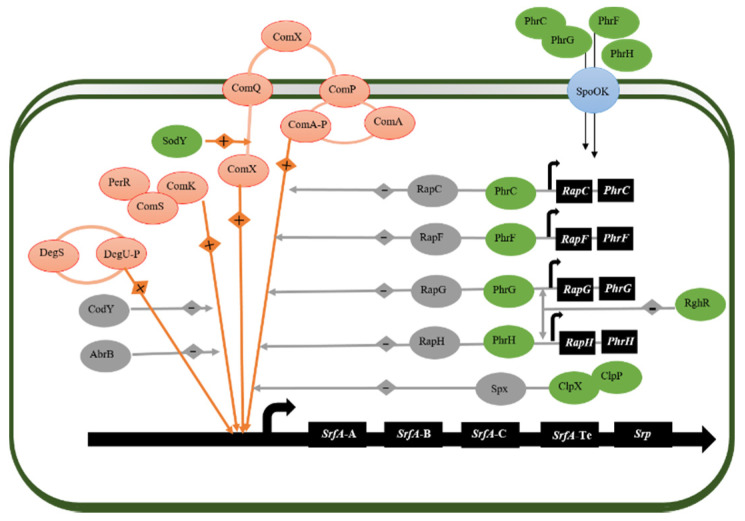
A gene regulatory model of surfacin biosynthesis. Orange indicates positive regulation and gray indicates negative regulation.

## Data Availability

Not applicable.

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
