# Peer review of "Microbial Lipopeptide-Producing Strains and Their Metabolic Roles under Anaerobic Conditions"

_microorganisms, 2021, doi:10.3390/microorganisms9102030_

Round 1
Reviewer 1 Report
This review manuscript summarizes bacteria which producing biosurfactants under anaerobic environment, and also general information concerning to the anaerobic metabolism and molecular mechanisms of NRPS. Although there is an industrial demand for microbial biosurfactants, most of them are produced in the aerobic environments. Reports on the biosurfactants production under anaerobic conditions, which are more useful for industrial applications, are few in number, but have been increasing in recent years, thus it is worthwhile to summarize them at this stage.
Chapter 2, "Lipopeptide producing strains under anaerobic conditions," is a comprehensive description of Bacillus species that produce biosurfactants under anaerobic conditions that have been reported so far. It is thought that this chapter is unique and most worthful part of this review.
Chapters 3 and 4 provide a brief overview of anaerobic metabolism and molecular mechanisms of NRPS of genus Bacillus. These topics are not limited to biosurfactants production but cover a wide range of physiological and ecological aspects, and many excellent and detailed reviews have already been published. There is no mention concerning to the relationship between biosurfactants and these topic in the text, nor there is an explanation of metabolic roles of biosurfactants as stated in the title.
For making this manuscript as valuable review, authors should focus on only biosurfactants-producing bacteria under anaerobic conditions. And. authors should enrich chapter 2 by describing microorganisms other than Bacillus species and providing chemical structure, activity, function, and other data.
Author Response
Point 1: Chapters 3 and 4 provide a brief overview of anaerobic metabolism and molecular mechanisms of NRPS of genus Bacillus. These topics are not limited to biosurfactants production but cover a wide range of physiological and ecological aspects, and many excellent and detailed reviews have already been published. There is no mention concerning to the relationship between biosurfactants and these topic in the text, nor there is an explanation of metabolic roles of biosurfactants as stated in the title.
Response 1: Thanks for your comments. The conclusions in the revised manuscript have been revised according to the review comments. The details are as follows:
This review summarized the thirty-six reported lipopeptide biosurfactants producing strains and twety-two other biosurfactants producing stains, mostly isolated from oil reservoirs. And the reported anaerobic biosurfactant-producing bacteria include thirty-seven strains of Bacillus, sixteen strains of Pseudomonas, one strain of Clostridium pasteurianum, two strains of Thermoanaerobacter, one Anaerophaga thermohalophila, one Geobacillus pallidus, one Rhodococcus ruber, one Tepidibacter mesophilus, one Luteimonas huabeiensis, one Enterobacter cloacae and one Desulfovibrio desulfuricans. Nitrate is an alternative electron acceptor for mainly strains under oxygen-limiting conditions, including Desulfovibrio and Thermoanaerobacter, which select sulfate as an alternative electron acceptor. This shows that bacteria is able to produce biosurfactants under anaerobic conditions, although the anaerobic biosurfactant yield much lower than that under aerobic conditions. The low anaerobic biosurfactant production of bacteria is contributed by the slow growth of bacteria in oxygen-limiting conditions. And the limited energy, producing in anaerobic conditions, is preferentially used in primary metabolic pathways rather than secondary metabolic pathways, which is also a major factor affecting the yield of biosurfactants. On the other hand, the difference of intercell microenvironments under oxygen-riched and oxygen-limited environments may affect the substrate perferences change due to the redox potential of the biochemical reactions and the usage of different electron acceptors. In general, the interaction of various factors leads to the low biosurfactants production of bacteria. Thus, undersdanding the factors affecting biosurfactant-produciion under microaerophilic and anaerobic conditions is important to the promising applications of there strains in bioremediation, microbial enhance oil recovery (MEOR), and the naturally oxygen-limiting environments. Most of the research on microbial enhance oil recover (MEOR) by biosurfac-tant-producing bacteria under anaerobic conditions was in the laboratory stage, and the in-situ microbial flooding experiments on MEOR by anaerobic bacteria was still scarce. It is of great significance for the research and application of anaerobic biosurfactant production technology independent of gas injection to screen the strain resources of anaerobic biosurfactant production from the reservoir and analyze its bio-surfactant production and metabolism process under the condition of oxygen deficiency.
In general, during the utilization of any anaerobically-produced biosurfactants strain for bioremediation or MEOR processes, the nature of used strain both from biosurfactants structure and the anaerobic growth and metabolic pathway must be examined first. Bacillus subtilis is most important of the anaerobic biosurfactant producer, the biosynthesis of surfactin is based on non-ribosomal peptide synthetases (encoded by srfA operon), which is regulated by quorum sensing system, a density-dependent signaling mechanism of microbial cells, and two-component signal transduction systems (TCS), using to couple changes in the extracellular environment to physiological effects. Bacillus subtilis relies on a set of internal complex regulatory network to make adjustments and choose the appropriate generation energy and metabolic pathway to cope with environmental changes. It is proved that the facultative anaerobe Bacillus subtilis may use nitrate replace oxygen as terminal electron acceptor to transform energy and maintain the oxidation and reduction potential equilibrium in cell under oxygen-limiting conditions. The pyruvate fermentation is also a way in the absence of available electron acceptor and oxygen. The energy required by Bacillus subtilis to synthesize lipopeptide can be obtained by nitrate respiration or pyruvate fermentation in oxygen limited environments. However, how the synthesis process of surfactin will change and adjust when the environmental oxygen level changes, and the specific impact degree and transformation strategy of Bacillus subtilis were unclear. Future research should intensify efforts in the difference of biosurfactants production of bacteria under the aerobic and anaerobic conditions. At the same time, the optimization of anaerobic biosurfactant production is also very effective for expanding the application range of anaerobic biosurfactant-producing Bacteria.
Point 2: For making this manuscript as valuable review, authors should focus on only biosurfactants-producing bacteria under anaerobic conditions. And. authors should enrich chapter 2 by describing microorganisms other than Bacillus species and providing chemical structure, activity, function, and other data.
Response 2: Thanks for your comments. The chapter 2 in the revised manuscript have been revised according to the review comments. The fifty-eight reported bacterial strains under anaerobic conditions were summarized. 22 strains of other biosurfactant-producing bacteria reported in the literature were added in table 2. However most of biosurfactants produced under oxygen-limiting conditions are still unidentified and uncharacterized, therefore, this review does not summarize the information of biosurfactant chemical structure, activity and function.
Table 2. Information of biosurfactant-producing strains under anaerobic conditions.
|
Year |
Source |
Strain |
Biosurfactant |
References |
|
1955 |
Ain-ez-Zania lake, Lybia |
Desulfovibrio desulfuricans sp. DSM 1926 |
Unidentified |
53 |
|
1991 |
Hot springs, USA |
Thermoanaerobacter pseudethanolicus ATCC 33233 |
Unidentified |
54 |
|
2000 |
Petroleum contaminated soil |
Pseudomonas sp.BS2201 |
Unidentified |
55 |
|
2000 |
Petroleum contaminated soil |
Pseudomonas sp. BS2203 |
Unidentified |
55 |
|
2000 |
Oilfield,India |
Pseudomonas aeruginosa ATCC 10145 |
Rhamnolipid |
56 |
|
2002 |
Oilfield; Hannover, Germany |
Isolate Glc2 |
Unidentified |
57 |
|
2007 |
Antarctic soil |
pantoa A-13 |
Rhamnolipid |
58 |
|
2007 |
Hospital Wastewater |
Pseudomonas aeruginosa PAO1 |
Rhamnolipid |
59 |
|
2008 |
Shengli oilfield,china |
Pseudomonas aeruginosa SH6 |
Rhamnolipid |
60 |
|
2010 |
Municipal Sewage Sludge |
Pseudomonas aeruginosa ANBIOSURF-1 |
Rhamnolipid |
61 |
|
2012 |
Oilfield; Veracruz, Mexico |
Thermoanaerobacter sp. |
Unidentified |
62 |
|
2012 |
Menggulin oilfield, China |
Pseudomonas aeruginosa WJ-1 |
Rhamnolipid |
63 |
|
2013 |
Soil samples, Iowa,USA |
Pseudomonas aeruginosa E03-40 |
Rhamnolipid |
64 |
|
2013 |
Oil contaminated soil, Iran |
Enterobacter cloacae |
Unidentified |
65 |
|
2014 |
Gachsaran oilfield, Iran |
Bacillus stearothermophilus SUCPM#14 |
Unidentified |
66 |
|
2015 |
Xinjiang oilfield,china |
Pseudomonas aeruginosa SG |
Rhamnolipid |
67-69 |
|
2017 |
Xinjiang oilfield,china |
Pseudomonas aeruginosa 709 |
Rhamnolipid |
44 |
|
2018 |
Daqing oilfield,china |
Pseudomonas aeruginosa DQ3 |
Rhamnolipid |
70 |
|
2018 |
Daqing oilfield,china |
Pseudomonas aeruginosa DQ1 |
Rhamnolipid |
70 |
|
2018 |
Daqing oilfield,china |
Pseudomonas aeruginosa DQ5 |
Rhamnolipid |
70 |
|
2018 |
Daqing oilfield,china |
Pseudomonas aeruginosa DQ6 |
Rhamnolipid |
70 |
|
2020 |
Dagang oilfield, china |
Bacillus licheniformis DM-1 |
Exopolysaccharide |
71 |
Reviewer 2 Report
The authors presented a review paper on the bacteria strains that have been reported to have the ability to produce lipopeptides under anaerobic conditions. The paper summarised the findings that have been published on the mechanisms of anaerobic growth of these bacteria.
The paper is quite to easy to follow and grammatically sound. Authors covered a sufficient range of relevant publications and made a good summary of the key findings. The paper as a review paper may facilitate people who are interested in relevant research.
The comment I have for the paper is mainly the conclusion and discussion, which is rather superficial. The purpose of the paper is to evaluate the industrial application of such kind of bacteria to produce lipopeptides given many applications are in anaerobic conditions. However, only Bacillus subtilis were discussed and covered in this review, which only takes roughly half of the bacteria strains that are anaerobic capable. They may employ quite different mechanisms and pathways, it would be good to compare them and that can make the study significantly more meaningful. The other thing is the paper can use a little more discussion to put everything together, e.g. what these mechanisms mean to industry and how may be applied.
Author Response
Point 1:The comment I have for the paper is mainly the conclusion and discussion, which is rather superficial.
Response 1: Thanks for your comments. The conclusions in the revised manuscript have been revised according to the review comments. The details are as follows:
This review summarized the thirty-six reported lipopeptide biosurfactants producing strains and twety-two other biosurfactants producing stains, mostly isolated from oil reservoirs. And the reported anaerobic biosurfactant-producing bacteria include thirty-seven strains of Bacillus, sixteen strains of Pseudomonas, one strain of Clostridium pasteurianum, two strains of Thermoanaerobacter, one Anaerophaga thermohalophila, one Geobacillus pallidus, one Rhodococcus ruber, one Tepidibacter mesophilus, one Luteimonas huabeiensis, one Enterobacter cloacae and one Desulfovibrio desulfuricans. Nitrate is an alternative electron acceptor for mainly strains under oxygen-limiting conditions, including Desulfovibrio and Thermoanaerobacter, which select sulfate as an alternative electron acceptor. This shows that bacteria is able to produce biosurfactants under anaerobic conditions, although the anaerobic biosurfactant yield much lower than that under aerobic conditions. The low anaerobic biosurfactant production of bacteria is contributed by the slow growth of bacteria in oxygen-limiting conditions. And the limited energy, producing in anaerobic conditions, is preferentially used in primary metabolic pathways rather than secondary metabolic pathways, which is also a major factor affecting the yield of biosurfactants. On the other hand, the difference of intercell microenvironments under oxygen-riched and oxygen-limited environments may affect the substrate perferences change due to the redox potential of the biochemical reactions and the usage of different electron acceptors. In general, the interaction of various factors leads to the low biosurfactants production of bacteria. Thus, undersdanding the factors affecting biosurfactant-produciion under microaerophilic and anaerobic conditions is important to the promising applications of there strains in bioremediation, microbial enhance oil recovery (MEOR), and the naturally oxygen-limiting environments. Most of the research on microbial enhance oil recover (MEOR) by biosurfac-tant-producing bacteria under anaerobic conditions was in the laboratory stage, and the in-situ microbial flooding experiments on MEOR by anaerobic bacteria was still scarce. It is of great significance for the research and application of anaerobic biosurfactant production technology independent of gas injection to screen the strain resources of anaerobic biosurfactant production from the reservoir and analyze its bio-surfactant production and metabolism process under the condition of oxygen deficiency.
In general, during the utilization of any anaerobically-produced biosurfactants strain for bioremediation or MEOR processes, the nature of used strain both from biosurfactants structure and the anaerobic growth and metabolic pathway must be examined first. Bacillus subtilis is most important of the anaerobic biosurfactant producer, the biosynthesis of surfactin is based on non-ribosomal peptide synthetases (encoded by srfA operon), which is regulated by quorum sensing system, a density-dependent signaling mechanism of microbial cells, and two-component signal transduction systems (TCS), using to couple changes in the extracellular environment to physiological effects. Bacillus subtilis relies on a set of internal complex regulatory network to make adjustments and choose the appropriate generation energy and metabolic pathway to cope with environmental changes. It is proved that the facultative anaerobe Bacillus subtilis may use nitrate replace oxygen as terminal electron acceptor to transform energy and maintain the oxidation and reduction potential equilibrium in cell under oxygen-limiting conditions. The pyruvate fermentation is also a way in the absence of available electron acceptor and oxygen. The energy required by Bacillus subtilis to synthesize lipopeptide can be obtained by nitrate respiration or pyruvate fermentation in oxygen limited environments. However, how the synthesis process of surfactin will change and adjust when the environmental oxygen level changes, and the specific impact degree and transformation strategy of Bacillus subtilis were unclear. Future research should intensify efforts in the difference of biosurfactants production of bacteria under the aerobic and anaerobic conditions. At the same time, the optimization of anaerobic biosurfactant production is also very effective for expanding the application range of anaerobic biosurfactant-producing Bacteria.
Point 2:The other thing is the paper can use a little more discussion to put everything together, e.g. what these mechanisms mean to industry and how may be applied.
Response 2: Thanks for your comments. The conclusions in the revised manuscript have been revised according to the review comments. The research on microbial enhance oil recover (MEOR) by biosurfactant-producing bacteria under anaerobic conditions was added. The details are as follows:
Most of the research on microbial enhance oil recover (MEOR) by biosurfactant-producing bacteria under anaerobic conditions was in the laboratory stage, and the in-situ microbial flooding experiments on MEOR by anaerobic bacteria was still scarce. It is of great sig-nificance for the research and application of anaerobic biosurfactant production tech-nology independent of gas injection to screen the strain resources of anaerobic biosur-factant production from the reservoir and analyze its bio-surfactant production and metabolism process under the condition of oxygen deficiency.
Round 2
Reviewer 1 Report
The points where I have mentioned were much improved. Biosurfactants producing bacteria except for genus Bacillus were also comprehensively summarized, and it enhanced the value of this review. In addition, the discussion part has also been detailed, it made that contents of Chapters 3 and 4 have become more important. I think that these improvements made that the title of the manuscript consistent with the body.
The text in the figures is difficult to see and understand, and needs to be improved to make it easier to read.
